# Modeling Visit Potential to Predict Hotspots of a Future District

**Younes Delhoum** and **Rachid Belaroussi** *

COSYS-GRETTIA, Université Gustave Eiffel, F-77447 Marne-la-Vallée, France; youndelh@hotmail.fr
* Correspondence: rachid.belaroussi@univ-eiffel.fr

**Abstract:** Understanding frequentation patterns allows urban planners to optimize the allocation of resources and infrastructure development. This includes determining the locations for schools, hospitals, public transportation, parks, and other amenities to efficiently meet the needs of the population. This paper proposes a study of the Visit Potential Model, an integrated model for evaluating the characteristics of public spaces. It is used to predict the potential potential presence of people in specific locations or public places. The model combines a universal law of visit frequencies in cities with a gravity measurement of accessibility. The adapted Visit Potential Model is represented as a graph by connecting public spaces to other spaces: population objects and attractor objects. Population objects represent places where people go in and out, such as houses, offices, and schools. Attractor objects include destinations that people visit, such as leisure parks and shopping malls. Originally, this static model was defined for a single time-frame by explicitly taking into the account the time component and a dynamic model was derived. A future district under construction was used as a case study: a multimodal transportation model was built to simulate and analyze the motion of people. The reported outcomes can be analyzed to provide us first insights of the potential for visiting the district's public spaces and define its future hotspots and places of interaction.

**Keywords:** daily planning; activity-based model; urban planning; city life; infrastructure

## 1. Introduction

Modeling the frequentation or visitation of places by people is crucial for various reasons. For urban planning and infrastructure development: understanding the frequentation patterns allows urban planners to optimize the allocation of resources and infrastructure development. This includes determining the locations for schools, hospitals, public transportation, parks, and other amenities to ensure they meet the needs of the population efficiently.

For transportation management: modeling visitation patterns helps in managing transportation systems effectively. By identifying peak times and popular routes, cities can optimize public transportation schedules and traffic flow to reduce congestion and improve overall transportation efficiency. Governments can use visitation data to evaluate the effectiveness of various policies, such as those related to transportation, housing, and urban development. It can help identify areas with unmet needs and inform policy adjustments. Modeling visitation patterns aids in optimizing the allocation of resources, both in urban and rural settings. This includes public funding distribution, emergency services deployment, and community support programs.

In business and economy: businesses heavily rely on visitation data to make informed decisions about their location strategies. Understanding foot traffic can help retail stores, restaurants, and other establishments identify prime locations, target specific customer segments, and optimize operating hours. It is the case especially in tourism planning that for tourist destinations, modeling visitation is critical in planning for crowd management, resource allocation, and creating enjoyable experiences for visitors. It can also help identify off-peak periods and develop strategies to promote tourism during slower seasons.

Other areas of interest are emergency response and healthcare. In times of emergencies or natural disasters, knowing where people are concentrated can be crucial for emergency response teams. Visit Potential Models can aid in allocating resources and providing timely assistance to affected areas. In the context of public health, modeling visitation can be used to track the spread of diseases and predict potential outbreaks. This data can help healthcare authorities respond effectively and implement targeted measures to control the spread of infections.

Visitation modeling is also useful in environmental impact assessment and social impact. Frequentation models can assist in assessing the environmental impact of human activities on specific areas, including identifying areas at risk of overuse and designing sustainable management strategies. Frequentation models can shed light on cultural and social interactions within a community, helping researchers and policymakers understand social behaviors, identify cultural hubs, and encourage community engagement.

## 2. Related works and Contribution

### 2.1. Literature Review

Urban vitality is a broad concept that encompasses a wide range of research, including socio-economic and demographic considerations, health, stress, quality of life, pollution, built environment, green and blue infrastructure, accessibility, safety, reliability, and resilience. We are interested in the question posed by urbanist Jane Jacobs [1]: what makes a city successful? The urban forms shaped by streets and buildings can either facilitate or inhibit social interactions. This research is based on the idea that a successful city must have a vibrant local life and meet the needs of various human activities. The concept of urban vitality is considered a fusion of information between predicting the intensity of public space use and the expected attractiveness of amenities to forecast future hotspots in a new planned neighborhood. This relies on modeling the typical daily activities of individuals, their movements, and real estate planning.

In this paper, we investigate the aspect of visits from an urban vitality point of view. In urban planning, urban vitality is defined as the level, in which the settlement pattern has supported vital functions and biological requirements [2]; Montgomery [3] defines urban vitality as activity, diversity, and transactions, where the people are on the streets all day, for different purposes and activities; and for Ravenscroft [4], urban vitality refers to how busy an urban center is, at different times and in different locations.

Several studies and works have been attempted to classify the indicators of urban vitality. Ref. [5] divided the indicators of urban vitality based on the level of significance into two groups. The indicators of first category of indicators are related to pedestrian flows; in contrast, the second set of indicators is mainly related to land use, security, accessibility, and parking facilities. In terms of performance indicators, Ref. [6] has classified the urban indicators of a city center into different classes, such as number of visits, demographic changes, safety, public transportation, and parking availability.

Maas [7] defined three main components of urban vitality: (1) the continuous presence of people in public spaces and streets, (2) the different activities and opportunities offered, and (3) the environment in which these activities took place.

In recent years, significant efforts have been devoted to implementation and, therefore, the application of indicators of urban vitality. These approaches address the fundamental factors of urban vitality: the built environment and human activity. A set of recent works are briefly presented in the following.

Gan et al. [8] investigated the relationship between block size and its impacts on the level of urban vitality, employing certain measures, for instance, the density of point of interest and social media check-in.

Li et al. [9] defined an indicator that estimates urban vitality at several levels. Social, economic, and cultural vitality were measured based on taxis and shared bicycles trajectories, user ratings, and points of interest data. Similarly to [9], Kim et al. [10] estimated the urban vitality at three dimensions: social, economic and virtual. Social vitality is measured

from the generated pedestrian traffic based on mobile phone activity; economic vitality is estimated using bank card transaction data, while the virtual vitality was calculated from the location of Wi-Fi hotspots.

Zeng et al. [11] defined an urban vitality index at the city and the sub-district levels; this vitality index is based on bike sharing data and it couples gravity-based accessibility measures with land-use mix index. Yue et al. [12] proposed a conceptual model for measuring urban vitality, which consists of indicators of: human activity, the built environment, and human–environment interactions.

Sulis et al. [13] proposed a computational approach to Jane Jacobs' concept of diversity and vitality [1], taking into account intensity, variability, and consistency—the three dynamic attributes of diversity. Vitality was estimated based on the flow of users in stations and stops.

Deprêtre et al. [14] are currently working on an vitality indicator that estimates urban intensity, based on City Information Model (CIM) data of the district of LaVallée. In this work, the authors approach the notion of urban intensity as an intensity of use of public spaces. The proposed indicator takes into account the time spent in a space, in addition to the very important potential of use, estimated from a set of attributes, for instance, environmental and demographic attributes.

Mouratadis and Poortinga [15] investigate the factors that impact urban vitality. They utilize a combination of population-based survey information and geospatial data. The geospatial data provides insights into key built environment features, including proximity to the city center via pedestrian routes, neighborhood density, accessibility to local amenities, availability of public transport, and the presence of green spaces. Meanwhile, urban vitality, social cohesion, socio-demographic attributes, and life satisfaction metrics were assessed through the population-based survey.

From the works cited, it can be concluded that the urban vitality is most often measured from two main dimensions, which are human activity and infrastructure [2,3,7]. Scientific studies typically incorporate built environmental measures (e.g., density, diversity, design), a human centered point of view, or both. Studies investigating the infrastructural layout usually measure urban vitality at the scale of the city or at a larger macroscopic scale. Studies related to human perception of places or computer vision based assessment is performed at a microscopic scale. The mesoscopic approaches are implemented at scale of the neighborhood; a recent survey can be found in [16].

The aspect of urban vitality investigated here is the popularity of a public place at the mesoscopic scale of the neighborhood. Human activity and foot traffic is of most importance and is the focus in this study. Urban hotspots are areas characterized by high levels of human movement, bustling foot traffic, and thriving economic endeavors. Typically situated in the city center or near prominent urban landmarks such as the main commercial street, central business district (CBD), and city square, these hotspots serve as the focal point of urban activities. They provide insights into the travel patterns and habits of people in different areas, as well as their tendency to spend time in specific locations.

Xu and Chen [17] used traditional field investigations, and pedestrian counting to measure the spatial features and population density: on-site headcount per unit of time and space is still practical, and while it is time-consuming, it can still be used in small-scale spaces.

Human centered methodology can assess the pedestrian level of service (PLOS) [18] which is a concept that comes from treating walking as a mode of transport to evaluate pedestrian traffic characteristics and network capacity (similar to vehicular LOS). Cepolina et al. [19] proposed a new methodology based on the individual level of comfort perceived by each pedestrian that moves in the area. At each time instant, each pedestrian perceives a comfort level based on the space they feel is currently available and the required space that is dependent on their walking direction as well as their physical and psychological factors. Ewing and Handy [20] conducted a study examining the relationship between twenty streetscape features and pedestrian traffic volumes. They identified three

key influential features—namely, transparency, active street frontage, and street furniture. Their findings suggested a strong correlation between pedestrian activity and the presence of street furniture, the proportion of active versus inactive land use, and the transparency of ground floor facades. In a separate study by Li et al. [21], computer vision technology was employed to assess foot traffic, which was then correlated with quantitative measurements of street built environment features. The study revealed that street width and transparency had significant positive effects on street vitality.

Tang et al. in [22] the willingness to stay in a space was evaluated by urban designers and compared with a computer vision algorithm. The algorithm assessed the visual quality of street space using Tencent Street View images, considering factors such as greenery, openness, enclosures, street wall continuity, and cross-sectional proportion. Human operators measured subjective criteria including enclosure, human-scale, transparency, tidiness, and imageability. Ewing et al. [23] added complexity to the measured attributes, but other type of indicator of urban ambiances and auditing public spaces can be found in [24,25]. These are important streetscape qualities for promoting population aggregation at a microscopic scale, and are influencing mechanisms of urban vitality. In [26], street quality, and environmental comfort (building continuity, greenness, openness, and walkability) are measured from street view images, and urban function refers to the residential, commercial, and public uses of an area; they are then linked to crowdedness information gathered from location-based services (LBS) data. They conclude the main influencing factors on vitality is the presence of facility conveniences and points of interest. These approaches are valuable for existing infrastructure in post-occupancy scenarios, but it is less suited for predicting the ambiance of future built environments.

A recent survey [27] depicts of how social media data are used to study places attachments and explore how people experience cities. Xia et al. [28] attempted to delineate the urban hotspot scope using taxi trajectory data with a network-based spatio-temporal clustering approach to discover and identify hotspots. Clustering of taxi GPS data is often performed for urban hotspot identification [29–31].

Others, such as Holleinstein and Purves [32], harvested georeferenced and tagged metadata associated with 8 million Flickr images to evaluate how large numbers of people name city core areas. Louail et al. [33] estimate the quantity of hotspots (defined as areas where mobile phone users gather) in relation to the total city population, from mobile phone data. They also noted the consistent spatial and temporal characteristics of these hotspots, signifying them as the central hubs of cities. In [34], population was estimated using the mobile phones data aggregated hourly, with land-use data; similarly, in [35], mobile phone data were aggregated over one day period and at the scale of Traffic Analysis Zones as the spatial units to define the neighborhoods. The authors of [36] also used mobile phone data for pedestrian volume modeling and prediction. The estimated pedestrian density from Monday to Thursday is used for pedestrian density prediction on Friday.

Wu et al. [37] asked participants to wear GPS tracking devices to provide a more realistic and precise description of pedestrian volume. Their share of out-of-home, non-work activities was used to define neighborhood vibrancy. Others such as Jiang et al. [38] used LBS data (mobile phones, bus cards, taxis data) to measure street activity intensity. They mapped them to streetscape quality assessed from street view images labeled by computer vision.

These approaches are inherently disintegrated: they assess different facets of public spaces, utilizing diverse methods to gauge human activity within them, and often focus on specific types of urban spaces or particular locations. Consequently, it is not always evident to what extent these findings can be extrapolated to inform the design of other public spaces. This is especially true for new designs that lack existing human activity for assessment. A more cohesive framework for quantitatively evaluating the qualities of public spaces would be advantageous. Such a framework should be applicable to all components of a network of public spaces and capable of modeling the potential presence of people in these areas.

*2.2. Contribution of the Research*

In addition to the works cited above, a particular work that caught our attention is an integrated model proposed by [39], which estimates the vitality of public spaces as the visit potential (VP). It combines a universal law of visit frequencies in cities [40] with a gravity measure of accessibility [41,42]. The defined model focuses on human activity, in particular, the pedestrian flows at the neighborhood scale.

This article pertains to the development of an urban characterization method based on activity-based travel models. We focus on modeling multimodal travel at the city scale, particularly micro-mobility. This has an original urban planning application within a mixed-use neighborhood construction project, where predictions are made about the vibrancy of the future neighborhood based on territorial digital traces and upcoming activities on the site. We focus on a peri-urban study area which is one of the gap between research and practice identified in the survey of Liu et al. [16] on neighborhood scale urban vitality assessment research. Also another gap of research identified is the multi-temporal aspect of hotspots in a neighborhood, where most of the research is focused on an aggregated one-day characterization. The existing body of literature predominantly emphasizes refining the evaluation methodology, with limited attention given to its applicability in contexts such as formulating action plans for local urban renewal initiatives.

What are the ingredients of a dynamic and vibrant city? Who uses the streets of a city and the surrounding areas, in what way, and for what reasons? These are essential questions in the realization of a new urban project. Indeed, the answers to these questions can not only help the planners of the project better anticipate the transportation needs in this area but also better prepare public spaces for potential uses they could support: vitality and attendance of the sites, preferred routes, and quality of public spaces. This article addresses the following research question: How to predict the urban hotspots of future real estate under development? One underlying question is how design and urban management impact vitality? We will base our method on dynamic transportation simulation and activity-based modeling for predicting human mobility and analysis of the amenities' popularity and frequency of visitors, using the current urban reference situation and the prospective scenario (i.e., when the district is built). This research addresses the increasing academic curiosity regarding the integration of systemic and scientific approaches into the traditionally qualitative and intuitive realms of urban design.

## 3. Materials and Methods

This article proposes an adaptation of the Visit Potential Model proposed in [39]. It is an integrated model for evaluating the characteristics of public space; this model combines a universal law of visit frequencies in cities [40] with a gravity measurement of accessibility.

To apply this model on a future district still under construction, we simulated the daily trips of a population of 126,000 agents of the five surrounding cities. We generated a synthetic population based on publicly available census data and aggregated zonal information travel survey [43].

The transportation simulation is an activity-based model with a dynamic traffic assignment; during the day an agent participates in different activities, with predetermined location and purpose, and the choice of departure time and mode of transport is estimated relative to traffic condition, personal routing behavior, and the individual possibility to adapt their hours [44]. In this study, agents with leisure and shopping trip purposes in their daily planning are proposed an alternate location inside the new district to conduct these specific activities, and we look for what factors can be determinant in a decision that triggers a change of location that favors visitation of the new district. These motions of people are used to apply the Visit Potential Model, especially the foot traffic at the level of the district.

Visit Potential Model is modeled as a graph by connecting public spaces to the other called spaces (objects). Objects are divided into three categories: population objects, attractor objects, and transport objects.

Population objects represent places where people going in and out, for instance, houses, offices, and schools. Attractor objects include destinations that people visit, such as leisure parks and shopping malls. Transport objects constitute the places where people change their mode of transport, such as bus stops and parking lots. For the sake of simplicity, the study explicitly focuses only on the population and attractor objects as illustrated in Table 1.

**Table 1.** Two types of object are used in this study.

| | |
|---|---|
| Population object | Home, Work, Education, Kindergarten |
| Attractor object | Shopping, Leisure, Restaurant |

The case study of this experiment is a future district under construction on 20 hectares in the outskirts of Paris, France, as illustrated by Figure 1. It is called LaVallée and its construction started in 2019 and the district is due to be completed in 2027. Working with the real estate company in charge of this new development in a suburban city, we have access to the infrastructure planning of the area. It was previously the location of a private school and except for the students and workers it was not used by the rest of the population. The whole district is to be delivered in 2027. Currently, the first roads and buildings are built and a first few inhabitants have settled in. This is Phase 1 out of 3 of the construction project. The district will provide the inhabitants with a new polarity of usages in the city with infrastructure of leisure, shopping, restaurants, schools changing it from a formerly closed area to a large public space of new activities. In this article, we are interested in the number of visitors the implantation of this new district will attract due to its leisure and shopping activities, both from Paris and from other cities around. Figure 1 shows the catchment area of the district which will attract visitors; it includes five cities with a total population of 126,000 residents. We built a dynamic traffic modeling (personal vehicle, public transport, walking, and bicycle) based on travel demand and infrastructure supply. The travel supply is made of the road network and facilities location. We build daily activity planning for a set of 126,000 agents (a synthetic population) and calculate the temporal evolution of planned trips. That is to say the predicted path of people during a typical day as well as their mode of travel over the area. The motion of people can then be used for urbanistic applications such as prediction of urban vitality, future hotspots, and places of interaction. Selecting the agents going inside the district, for purposes such as work, school, kindergarten, leisure, shopping, or restaurant, we can predict the ingress/egress rate to all facilities within LaVallée.

The forecasting Visit Potential Model is applied to the case study of LaVallée neighborhood. The reported outcomes can be analyzed to provide us first insights of the potential for visiting LaVallée's public spaces. Originally, this static model was defined for a single time-frame; by explicitly taking into the account the time component, a dynamic model can be derived.

Herthogs et al. [39] proposed three variants to predict the visit potential of public spaces:

- Visit Potential as Proximity to People $VP(1)$: it estimates the accessibility of a public space.
- Visit Potential as Accessibility of Attractors $VP(2)$: it estimates the potential number of people visiting attractors.
- Visit Potential as Aggregate Pedestrian Movement $VP(3)$: it estimates the number of people moving through and occupying a public space.

In the following, the three variants of Visit Potential Model are defined.

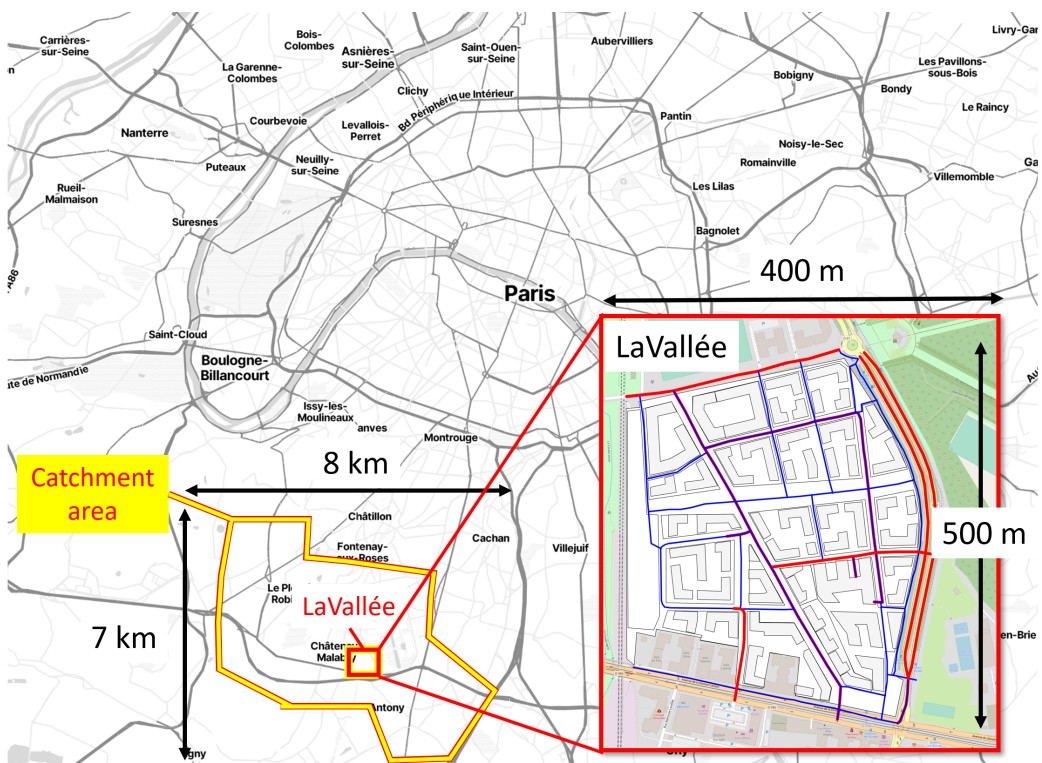

**Figure 1.** LaVallée district: a real estate project in the South of Paris. Transportation was simulated for 126,000 agents in the catchment area of five cities surrounding the district.

### 3.1. Visit Potential as Proximity to People VP (1)

The first variant $VP(1)$ estimates the accessibility to a public space, it is expressed as a gravity measure, and equal to the aggregate of attractiveness of the population objects modified by a distance decay factor. From a mathematical point of view, the visit potential as proximity to people of a public space $i$ is expressed in Equation (1).

$$VP(1)_i = \sum_{j=0}^{|J|} \frac{P_j}{(d_{ij})^2} \tag{1}$$

where

- $i$: a public space.
- $I$: set of public spaces ($i \in I$).
- $j$: a population object.
- $J$: set of population objects ($i \in J$).
- $P_j$: entrance/leaving rate of a population object $j$.
- $d_{ij}$: walking distance of the shortest path between $i$ and $j$.

### 3.2. Visit Potential as Accessibility of Attractors VP (2)

The second variant $VP(2)$ estimates the potential number of visitors to an attractor, the attractor has two main attributes: need frequency and maximum capacity. Need frequency refers to how often an agent visits an attractor, while maximum capacity represents the maximum number of people served by an attractor over a given time period. The potential number of visitors to the attractors of a public space $i$ is expressed in Equation (2).

$$VP(2)_i = \sum_{j=0}^{|J|} \sum_{a \in A_i} nf_a \cdot \frac{P_j}{(d_{ij})^2} \tag{2}$$

where:

- $A_i$: set of attractors which are connected to a public space $i$.
- $a$: an attractor.
- $nf_a$: need frequency of the attractor $a$.

For simplicity, the need frequency of a public space $i$ is given by Equation (3).

$$nf_i = \sum_{a \in A_i} nf_a \tag{3}$$

$VP(2)$ presents several drawbacks: on the one hand, the fact that a public space without an attractor will have no visit potential; on the other hand, this variant does not take into account the fact that a person who visits a specific attractor will also visit all the public spaces he passes through on the way.

### 3.3. Visit Potential as Aggregate Pedestrian Movement VP (3)

The third variant of VPM estimates the number of people moving through an occupying public space. In this variant, it is considered that all the people of a population object $j$ go to a public space $r$ with attractors; in order to determine the number of visitors of $r$, the entrance/leaving rate of a population object $P_j$ is split according to two weights: the first concerns the frequency of need public space attractors; the second is related to the inverse of the square of the distance between the population object and the public space.

The relative proportion of people going from population object $j$ to visit a public space with attractors $r$ is given by Equation (4).

$$w_{jr} = \frac{nf_r}{2\sum_{r=0}^{|R|} nf_r} + \frac{(d_{jr})^{-2}}{2\sum_{r=0}^{|R|} (d_{jr})^{-2}} \tag{4}$$

where

- $r$: a public space with attractors.
- $R$: set of public spaces with attractors ($R \subseteq I$).
- $nf_r$: need frequency of the set of attractors connected to $r$.

The number of people moving through an occupying public space $i$ is expressed in Equation (5).

$$VP(3)_i = \sum_{r=0}^{|R|} \sum_{j=0}^{|J|} SP_{jr,i} \cdot P_j \cdot w_{jr} \tag{5}$$

where

- $SP_{jr}$: set of public spaces that belong to the shortest path from $j$ and $r$.
- $SP_{jr,i}$: 1 if $i \in SP_{jr}$; 0 if $i \notin SP_{jr}$.

For more details on the different variants of the Visit Potential Model, we refer the reader to the corresponding work [39].

### 3.4. Dynamic Visit Potential Model

The initial variants of Visit Potential Model estimate the presence of people in a single time-frame, the presented variants can be transformed into dynamic models by including the time component. Therefore, $P$ is replaced by $P^t$, the rate of entrance/leaving of a population object for a time period $t$. Those dynamic variants are updated in the following.

First, the dynamic Visit Potential as Proximity to People is expressed in Equation (6).

$$VP(1)_i^t = \sum_{j=0}^{|J|} \frac{P_j^t}{(d_{ij})^2} \tag{6}$$

Second, the dynamic Visit Potential as Accessibility of Attractors is expressed in Equation (7).

$$VP(2)_i^t = \sum_{j=0}^{|J|} nf_i \cdot \frac{P_j^t}{(d_{ij})^2} \tag{7}$$

Third, the dynamic Visit Potential as Aggregate Pedestrian Movement is expressed in Equation (8).

$$VP(3)_i^t = \sum_{r=0}^{|R|} \sum_{j=0}^{|J|} SP_{jr,i} \cdot P_j^t \cdot w_{jr} \tag{8}$$

## 4. Case Study: Application of Visit Potential Model at LaVallée

We present the work on a district under construction called LaVallée. It is located in the south suburbs of Paris, as illustrated in Figure 1. A multimodal simulation of transportation was run; it takes into account people residing the future premises. The current inhabitant of a 15 min catchment area including five cities around the district and future worker originating from outside this area of influence shown in yellow in Figure 1. The methodology to generate activity plans for the population within the catchment area can be found in [44]. This population of 126,000 agents encompasses residents, externals (workers and students), and other potential visitors to the LaVallée district.

### 4.1. Visit Potential Model Objects

The whole LaVallée district is expected to be completes in 2027. Currently, the first roads and buildings are being built and the first inhabitants are expected by the end of 2023. The district will provide the inhabitants with a new polarity of usages in the city with the infrastructure of leisure, shopping, restaurants, and schools—transforming the area from a formerly closed area to a large public space of new activities.

The Visit Potential Model has two main types of objects, which are population and attractor objects. We refer the population objects to the facilities of the primary activities, which consist mainly of home, work, and education.

The rate of entrance/leaving $P_j$ of a population object $j$ is the relevant attribute, and it is defined as follows:

- $P_{home}$: represents the number of residents in the building.
- $P_{work}$: represents to the number of employees.
- $P_{education}$: refers to the number of students in the school.
- $P_{kindergarten}$: refers to the maximum number of children in the nursery.

The LaVallée population objects with entrance/leaving rates are shown in Figure 2.

In this study, the attractor objects are referred to facilities for secondary activities, for instance, shopping, leisure, and restaurant. As we saw in Section 3.2, an attractor has two attributes: maximum capacity and need frequency.

Attractors are considered oversized relative to the number of people present in the neighborhood. Thus, the impact of the maximum capacity of the attractor is not taken into account. The second attribute is the need frequency; it corresponds to the share of the trip purpose, which is defined as the proportion of travelers that perform an activity per day. Therefore, we propose to substitute the need frequency of an attractor by its share of trip purpose.

In the following, a specific process is applied to define the need frequency for attractors based on the data from national travel survey (Enquête Global de Transport EGT2010 [45]), this process is performed in three stages.

In the first stage, the share of trip purpose of the secondary activities, is extracted from the EGT2010 census.

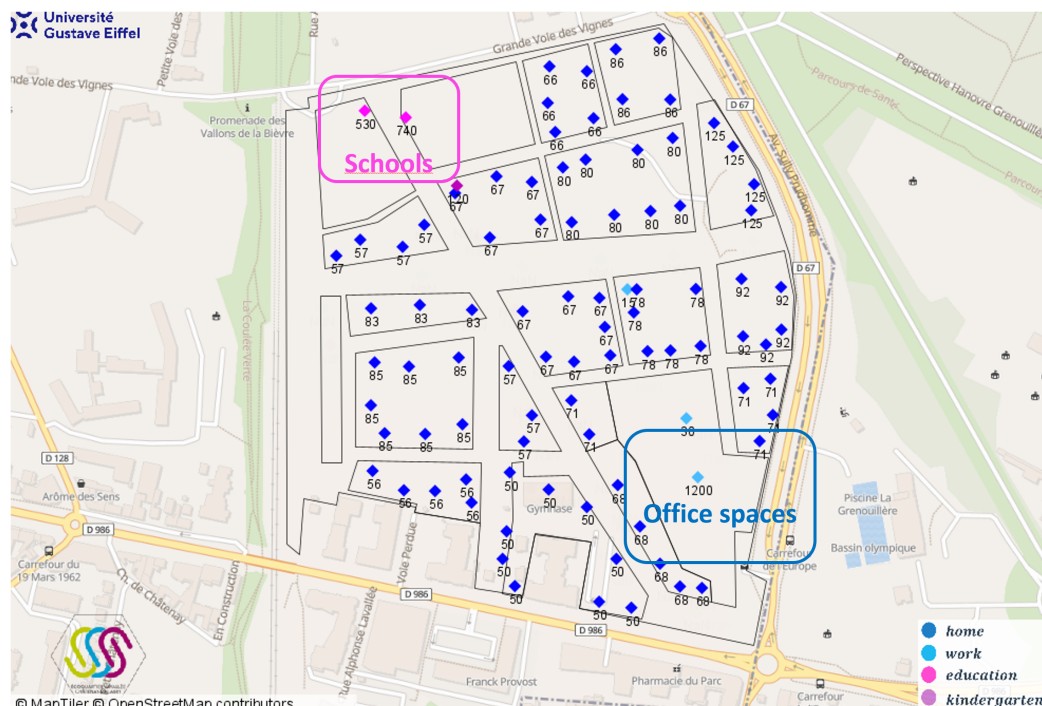

**Figure 2.** Population objects in LaVallée. A population object is annotated with its entrance/leaving rate, which refers to its defined capacity. For instance, a work object is annotated with its number of employees.

In the second stage, it is implicitly assumed that attractors of the same type have an equitable need frequency value. Therefore, the need frequency of an attractor object *j* is equal to the defined need frequency $nf_{type}$ divided by the number of attractors of *type*. The need frequency for the different attractors is presented in Table 2.

**Table 2.** Need frequency of the different type of attractors in LaVallée.

| Attractor *Type* | $nf_{type}$ | $|A_{type}|$ | $nf_j$ |
|---|---|---|---|
| Leisure | 17% | 9 | 1.9% |
| Shopping | 13% | 16 | 0.8% |
| Restaurant | 7% | 9 | 0.8% |

Figure 3 depicts the spatial distribution of attractor objects in the LaVallée district.

In this third stage of the given process, the need frequency for public spaces can be deduced based on Equation (3). Based on this last measurement, public spaces in LaVallée are grouped into four classes:

- *High nf*: public spaces belong to this class and are connected to multiple attractors. They are spatially located south of *Cours du Commerce*.
- *Medium nf*: they represent the smallest set of public spaces. On average, they are immediately accessible from two attractors.
- *Low nf*: this category includes the public spaces of *La Promenade Plantée*. Moreover, these spaces are connected to a single attractor.
- *Without nf*: this category covers public spaces without attractors.

The public spaces with a positive need frequency value are depicted in Figure 4.

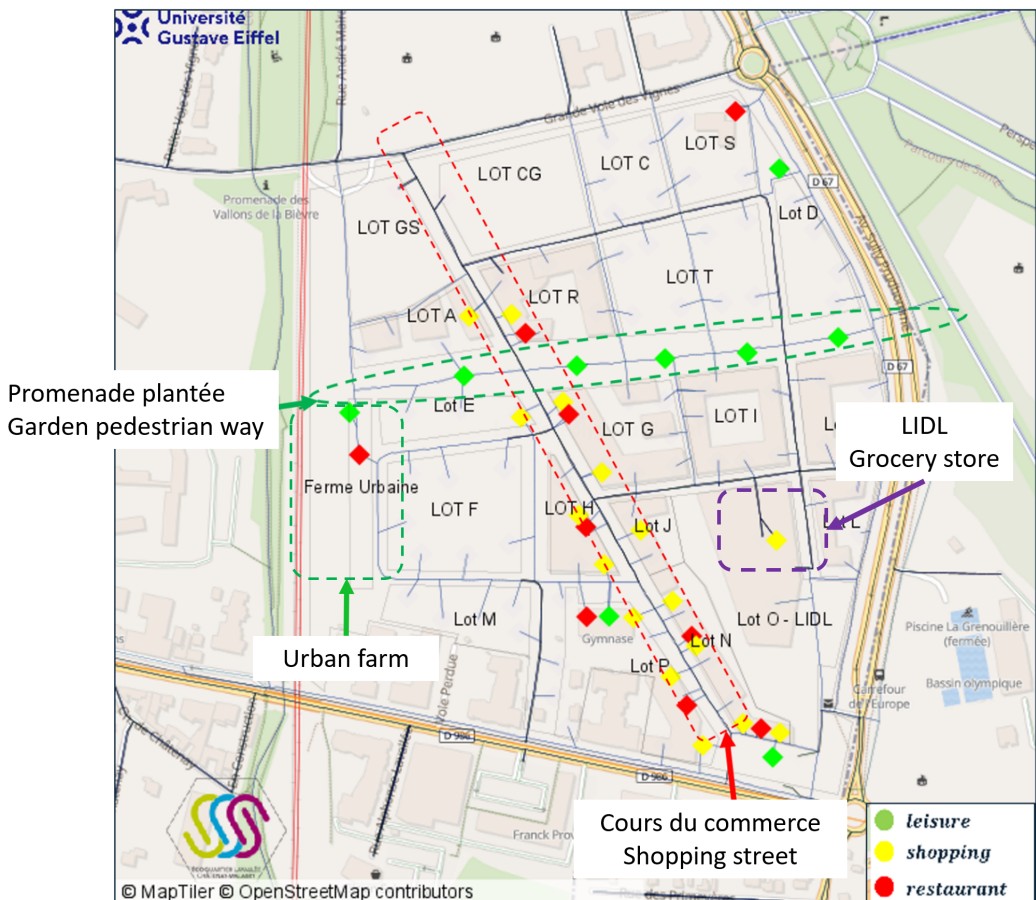

**Figure 3.** The spatial distribution of attractor objects in LaVallée. Attractors are categorized into three classes: leisure objects (green) are mainly located in *La Promenade Plantée*. Shops (yellow) and restaurants (red) are mainly situated along *Cours du Commerce*

### 4.2. Dynamic Entrance/Leaving Rate of Population Type

In this article, we are interested in the number of visitors the implantation of this new district will attract due to its leisure and shopping activities from Paris and other cities. To model the daily trips of people in the area, we used MATSim platform [46].

The work carried out addresses dynamic traffic modeling (road, public transport, and active modes) based on travel demand and infrastructure supply. The travel supply is made of the road network and facilities location. We build daily activity planning for a set of agents (a synthetic population), and calculate the temporal evolution of planned trips; that is to say that the predicted path of people during a typical day as well as their mode of travel [44].

We analyzed mobility patterns over a sub-regional area large enough to take into account all potential visitors. That region includes all inhabitants able to access the district in 15 minutes no matter the mode of transportation, taking into account restrictions related to the built environment. Elements structuring population mobility have been integrated such as the presence of a highway on the south, which acts as a barrier on possibility of trips toward or out the city. The planned tramway line is also a structuring element in the future mobility pattern. A total of 126,000 people were selected to constitute the agents of a synthetic population representing the resident of this area.

We used MATSim (Multi-Agent Transport Simulation [46]) framework for demand-modeling and agent-based mobility-simulation. It is open-source and widely used to implement large-scale activity-based modeling. Therefore, the method to build the daily planning (chain of activities) of each agent as well as the location and time of departure is described in [44,47]. Trip purposes included in this study are work, education, shopping,

leisure (including restaurant), and kindergarten. Transportation modes are private vehicle (car), public transportation, walk, and bicycle [48]. Public transportation is made of three modes: buses, which are assigned on the car network; railway; and tramways (with only two lines: the current Tram T6 and eventually Tram T10).

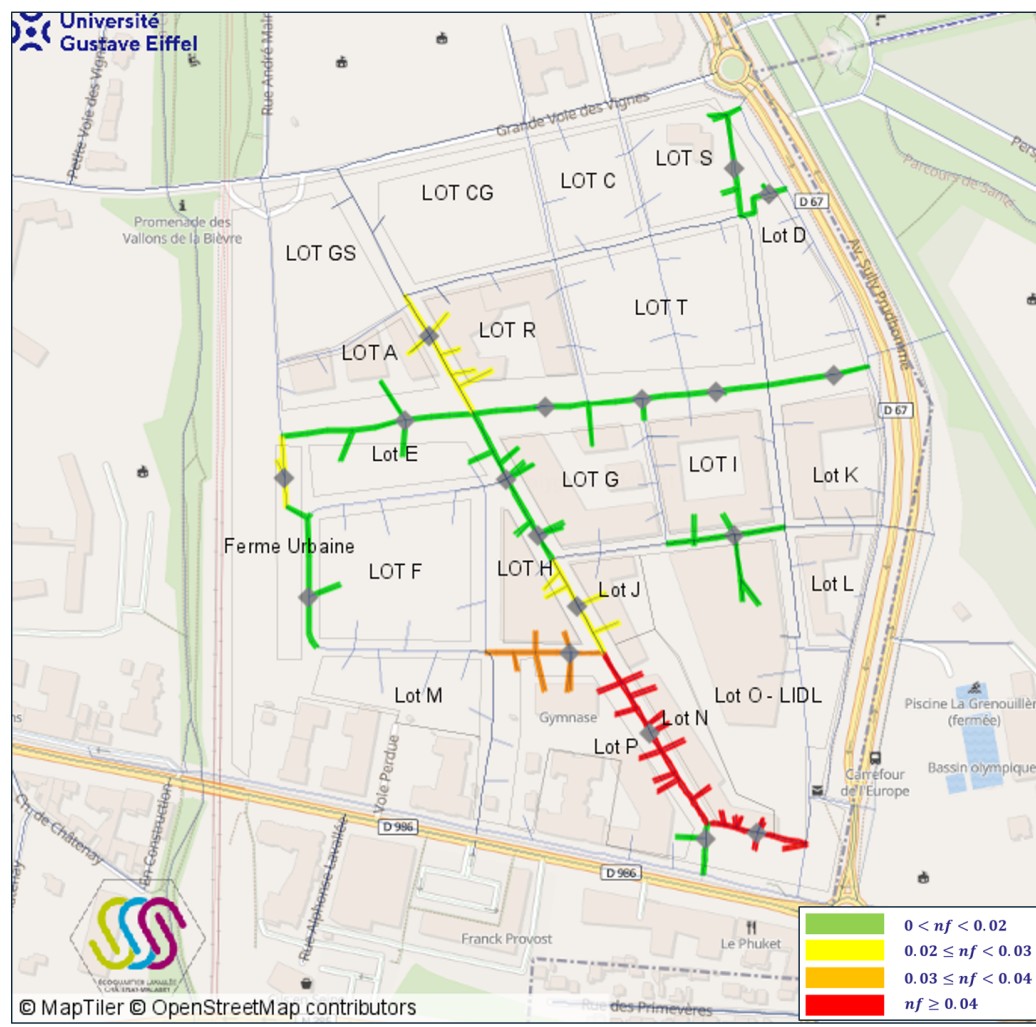

**Figure 4.** LaVallée's public spaces can be divided into four groups. Public spaces in red and orange have a high need frequency. Groups of public spaces in yellow and green designate spaces with medium and low need frequency, respectively. The public spaces without color have no *nf*.

The transportation simulation is an activity-based model with dynamic traffic assignment. During the day an agent participates in different activities, with predetermined location and purpose, and the choice of departure time and mode of transport is estimated relative to traffic condition, personal routing behavior, and the individual possibility to adapt their hours. In this study, agents with leisure and shopping trip purposes in their daily planning are proposed an alternate locations inside the new district to conduct these specific activities, and we look for what factors can be determinant in a decision that triggers a change of location that favors visitation of the new district.

From these motions of people we can define the dynamic entrance/leaving rate of a population object $p_j^t$. Entering a population object, for instance, work, corresponds to arrival at the workplace. The time span between the arrival at the place of the activity and the start of the activity itself, is relatively short and neglected compared to its duration.

According to the previous assumptions, the entrance time into a place of activity is equal to its start time. Correspondingly, the leaving time of an activity equals to its end time.

It can be concluded that for a period $t$, the dynamic entrance/leaving rate of a population object $j$ corresponds to its activity starting/ending rate.

Based on the provided results of the simulated scenario in the reference situation, the start/end rates of the different activities are the outcome of a specific process application. In essence, this algorithm extracts the ratio of agents who starting/ending an activity at a given time period $t$.

In technical terms, the proposed process is executed for each primary activity $act$ and for each time period $t$. This process consists of three steps.

First, the frequency of agents $sF_{act}^t$ who start an activity $act$ is determined from the simulation results provided for each period $t$, its relative frequency $sN_{act}^t$ is calculated as the ratio of $sF_{act}^t$ to $F_{act}$, which represents the total number of agents who perform $act$ during $t$. Similarly, the relative frequency of agents $eN_{act}^t$ that complete an activity $act$ during $t$ is then estimated.

Second, we can denote the global relative frequency $N_{act}^t$ as the sum of the two relative frequencies $sN_{act}^t$ and $eN_{act}^t$.

Third, the dynamic entrance/leaving rate $P_j^t$ is derived as the product of $P_j$ and $N_{act}^t$, this rate is expressed in Equation (9).

$$P_j^t = P_j \cdot N_{act}^t \tag{9}$$

The proposed process for calculating dynamic entrance/leaving rates is illustrated in Figure 5.

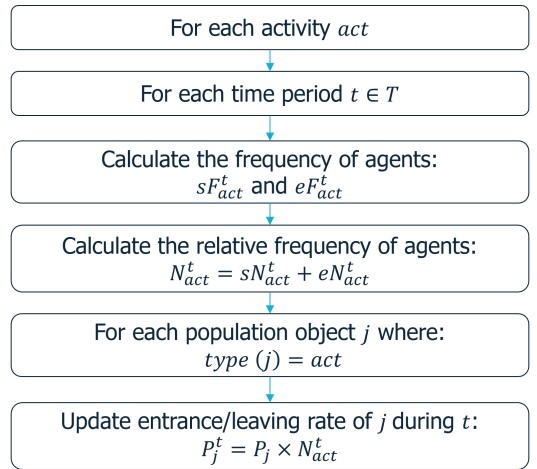

**Figure 5.** An overview of the process for calculating the dynamic entrance/leaving rates

Table 3 summarizes the frequency of agents performing an activity $act$ in the reference situation.

**Table 3.** Total number of agents carrying out an activity (in the reference situation).

| Activity $act$ | $F_{act}$ |
|:---:|:---:|
| Home | 126,151 |
| Work | 63,827 |
| Education | 26,271 |
| Kindergarten | 8971 |

Next, we denote by $T$ the set of one-hour periods included in the time range from 06h00 to 23h00. An illustration of the proposed algorithm on work activity is given in the following.

First, the frequency of agents $sF^t_{work}$ and $eF^t_{work}$ for each period $t \in T$ is shown side by side in Figure 6.

Second, the relative frequency of agents $sN^t_{work}$, $eN^t_{work}$, and $N^t_{work}$, for each time period $t \in T$ are presented in Figure 7.

Finally, by applying Equation (9), the dynamic entrance/leaving rate for work objects, for instance, *LIDL-headquarters* can be inferred (see Figure 8).

Figure 9 depicts the start/end rates of the different primary activities. Further down, the potential number of agents present in public spaces per one-hour period is shown in Figure 10.

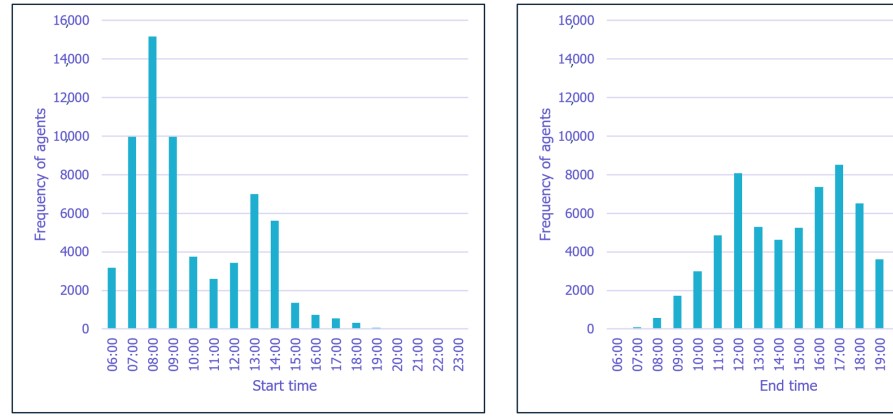

**Figure 6.** (**Left**): $sF_{work}$ frequency of agents starting work by time of day. (**Right**): $eF_{work}$ the frequency of agents who complete their work according to the time of day.

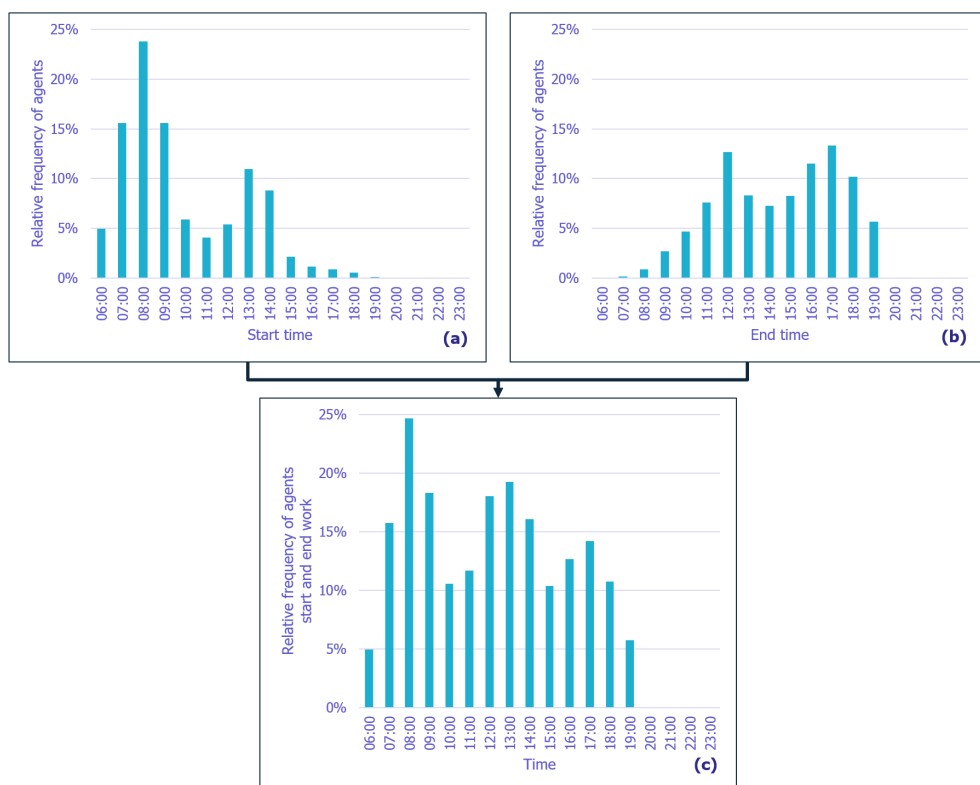

**Figure 7.** (**a**): $sN_{work}$ the relative frequency of agents starting work by time of day. (**b**): $eN_{work}$ the relative frequency of agents finishing working by time of day. (**c**): $sN_{work}$ total relative frequency of agents starting/ending work per time period.

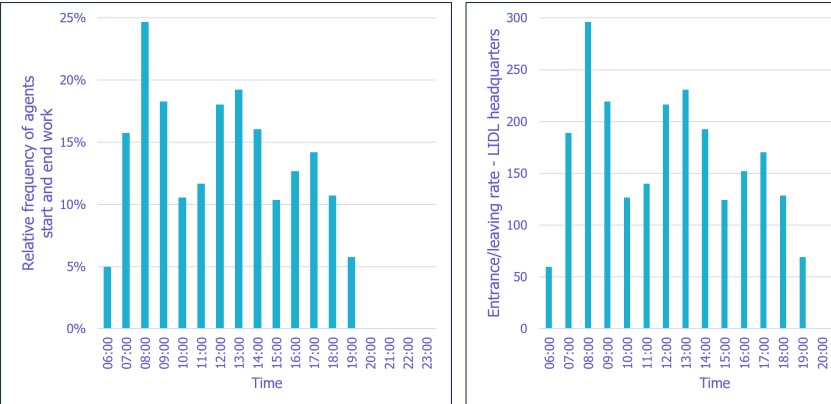

**Figure 8.** (**Left**): total relative frequency of agents starting/ending work per time period. (**Right**): *LIDL-headquarters* entrance and leaving rate between 06h00 and 20h00.

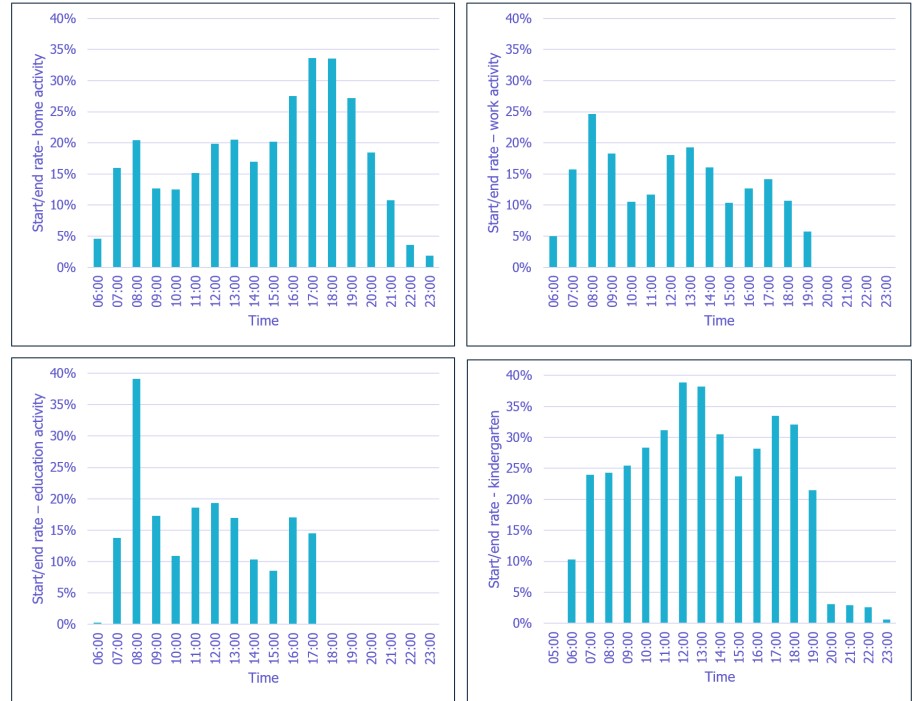

**Figure 9.** Rate of start/end of primary activities: home, work, education, and kindergarten.

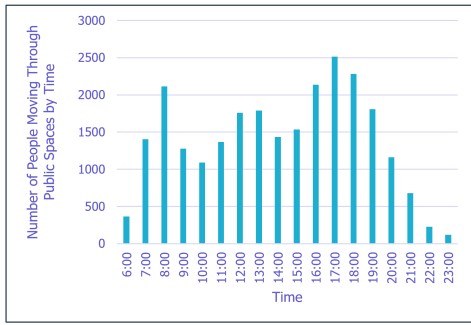

**Figure 10.** Potential number of people moving through public spaces per time period.

## 5. Results and Discussions

After having defined the population objects of LaVallée, their entrance/leaving rates, the attractor objects, and their need frequency, then the static and dynamic VPM variants can calculated and analyzed afterwards.

In what follows, the results of the application of *VPM* for a single time slot (06h00–23h00) are detailed and therefore discussed. Next, the results of the third dynamic variant $VP$ (3) are explicitly analyzed, with an illustration in three peak hours: morning, noon, and evening.

### 5.1. Visit Potential Model

The first variant $VP(1)$, estimates the visit potential in public spaces according to the proximity to population objects.

The results outlined in Figure 11 provide an overview of the visit potential of public spaces in LaVallée. According to the *VP* value, one can distinguish four different classes of public spaces, which are given as follows:

A first class with a very high *VP* (red color), contains solely the public space situated between *LOT GS* and *LOT CG*. This high value of visit potential is the consequence of the immediate proximity of this public space to the two schools (Groupe Scolaire and Collège) with a very high entrance/leaving rate.

A second class with a high *VP* (orange color), groups public spaces close to places of work and education, such as: LIDL-headquarters, Groupe Scolaire and Collège.

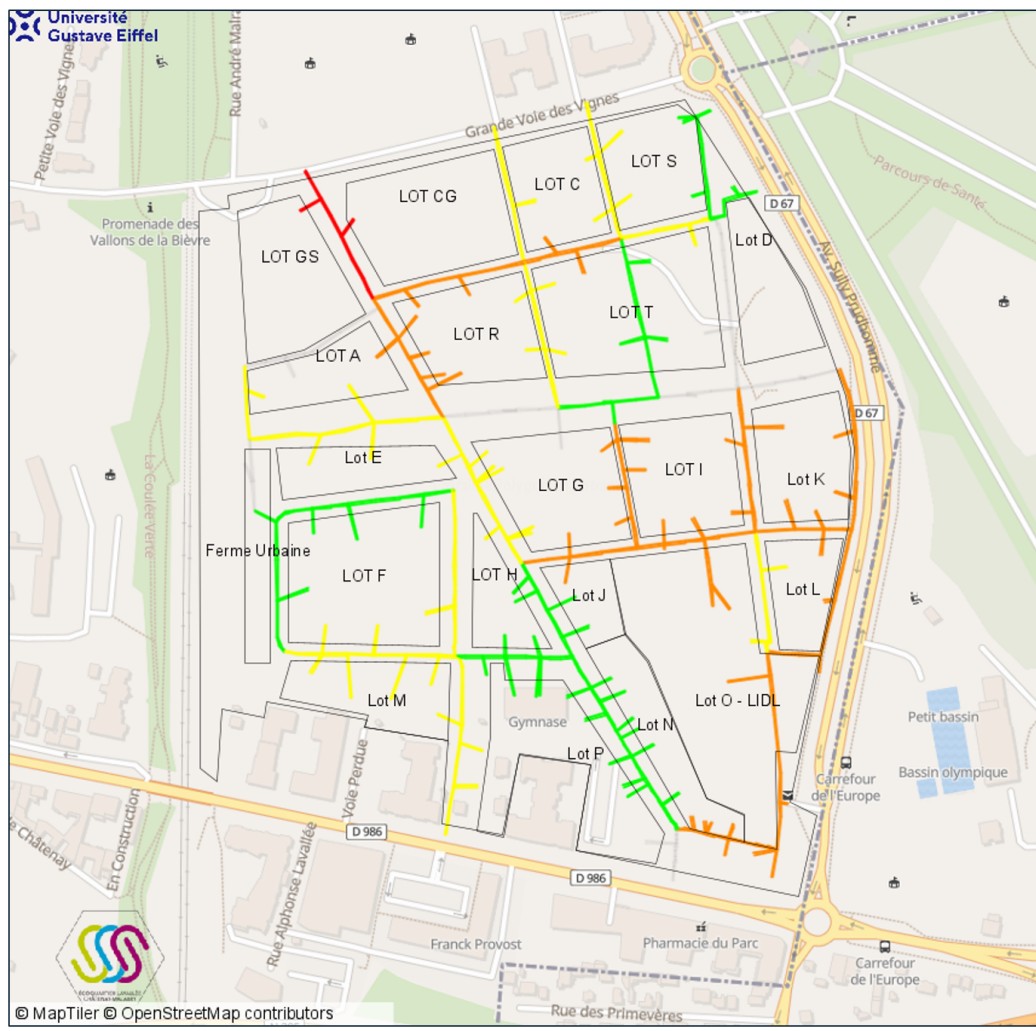

**Figure 11.** Visit Potential as Proximity to People in LaVallée. (Red): refers to a very high *VP*. (Orange): public spaces near the work and education objects, have a high *VP*. (Yellow): public spaces with medium *VP* and connected to multiple parcels. (Green): they are isolated or without significant access to population objects.

The third set with a medium *VP* value (yellow color) is notably made up of these public spaces which are connected to several parcels, for instance: *Lot C*, *Lot S*, and *Lot M*. These parcels include a large number of population objects, in particular residences.

The fourth group (green color) covers a set of public spaces; they are dispersed and relativity far from the dense population objects. Mainly, they are located at the edge of the district, for instance, *Ferme Urbaine*, *Lot D*, and *Lot P*. These spaces have a lower VP value compared to those in the center of LaVallée.

The second variant $VP(2)$, estimates the visit potential based on the accessibility to attractors. As can be seen in Figure 12, LaVallée's public spaces can be divided according to $VP(2)$ into three groups:

The public spaces of the first group (colored red and orange) have a high visit potential and can be divided into two separated clusters.

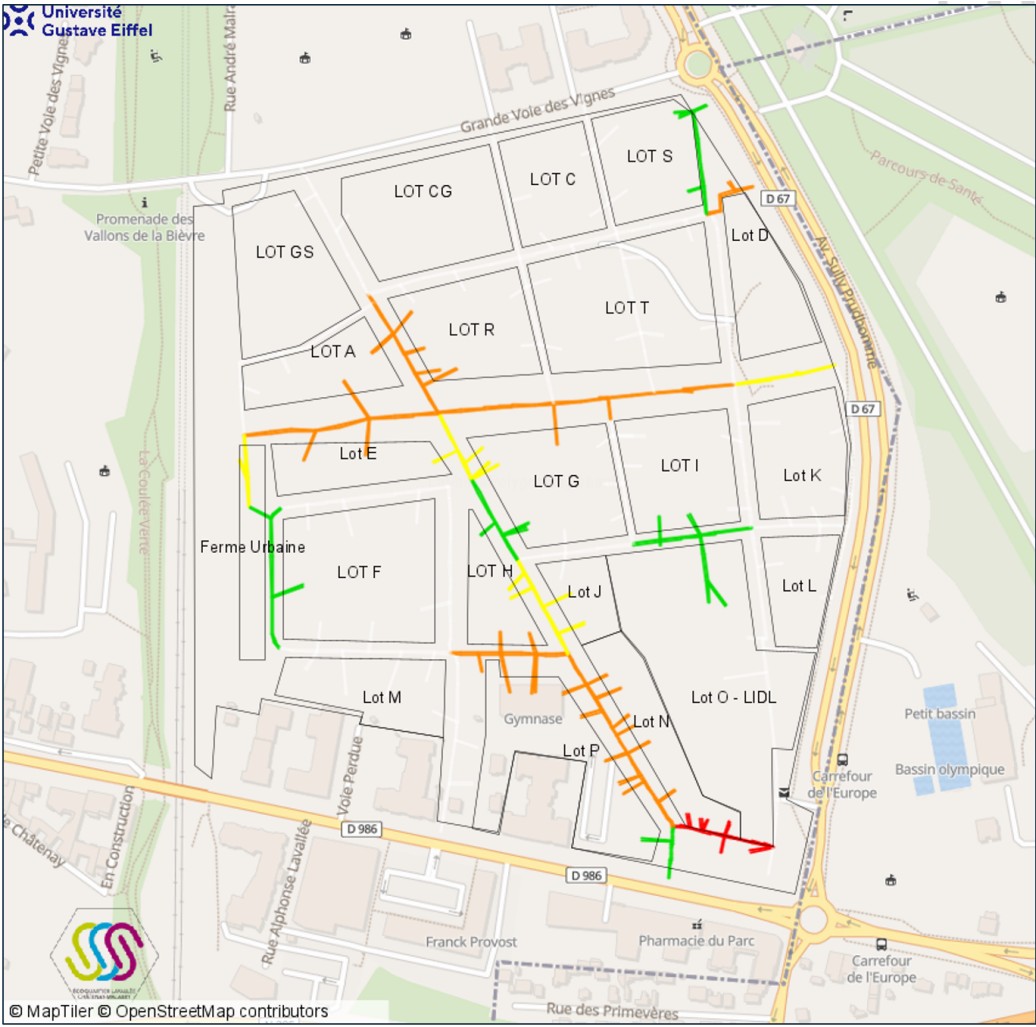

**Figure 12.** Visit Potential as Accessibility of Attractors in LaVallée. Public spaces colored: in red and orange have a high *VP* and are located along *La Promenade Plantée* and south of *Cours du Commerce*; in yellow and green have a medium *VP* and are situated far from dense population objects; and those that have no color, are without attractors and therefore a zero *VP*

- The first cluster brings together the public spaces that are in *Promenade Plantée*; they host a large number of leisure facilities and are immediately accessible to as many inhabitants.
- The second cluster is located south of *Cours du Commerce*, these public spaces are nearby *LIDL-headquarters*, which has the highest ingress/egress rate. In addition, this set of public spaces is home to numerous shops and restaurants. An exception is the

public space between *Lot D* and *Lot S*; it is connected to a leisure area and a restaurant and surrounded by several residential objects.

The elements of the second group (colored yellow and green), have a medium value of *VP*; they are generally connected to a single attractor and essentially situated at the limits of the neighborhood. Therefore, they are relatively far from a large number of population objects.

The third group gathers the public spaces whose visit potential visit value is null and are not connected to any attractor.

*VP (3)*, the third variant of *VPM*, estimates the visit potential based on the aggregated flow of pedestrians through public spaces. This variant is considered the most interesting *VPM*. From Figure 13, three sets of public spaces can be observed.

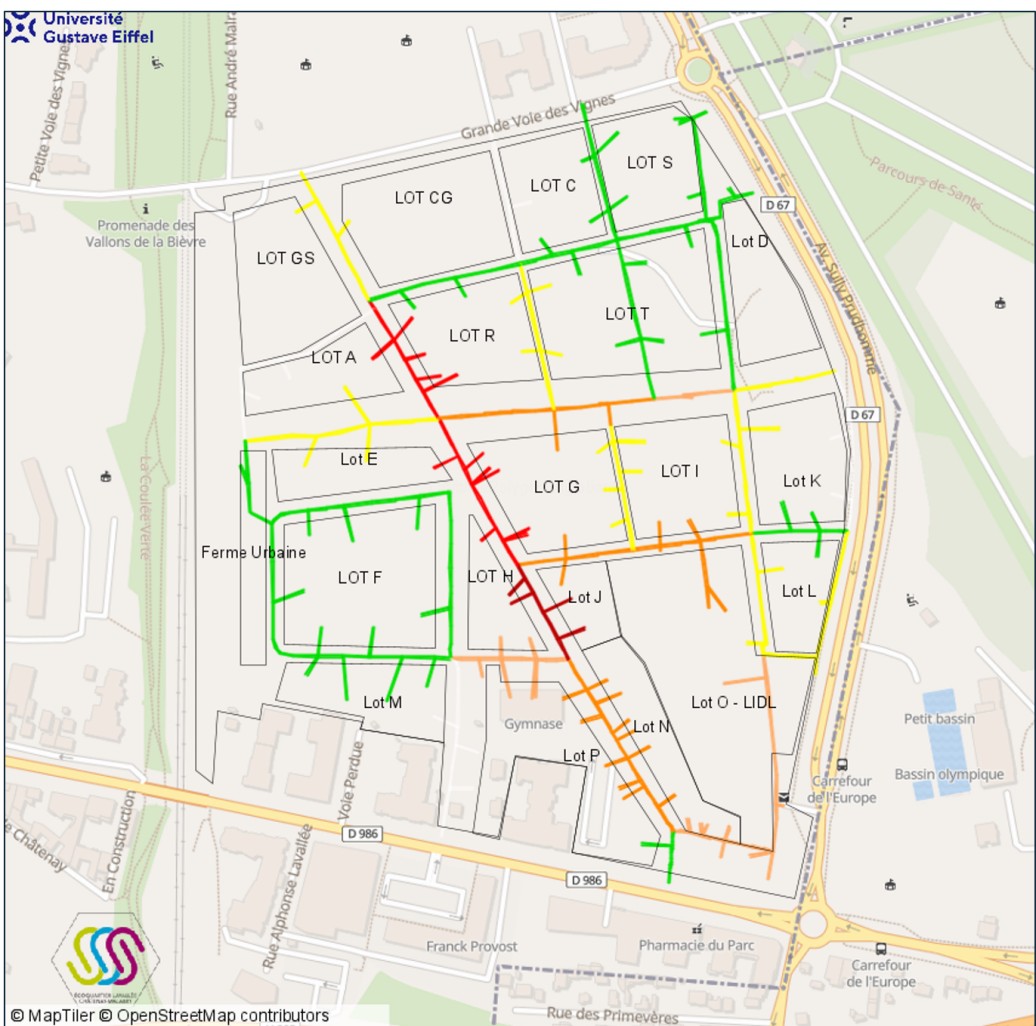

**Figure 13.** Visit Potential as Aggregate Pedestrian Movement in LaVallée. Public spaces colored: in red and orange, have a very high *VP* and are located near major population objects and several attractors; in yellow, have a moderate *VP*, they are without attractors but connected to very potential public spaces; and in green, they have a low *VP* and they are relatively far from the LaVallée center attractors.

The first cluster (colored in red and orange) is characterized by the high *VP* value of its public spaces, in particular those located in *Cours du Commerce*, *Promenade Plantée*, and around *LIDL-headquarters*. This high potential visit value is explained by the presence of several attractors and by the proximity of many population objects.

The public spaces of the second group (colored in yellow) have an average *VP* value. In fact, these public spaces are devoid of attractors, but they are immediately connected to public spaces with attractors.

The third set (in green color) is made up of public spaces with a low visit potential. They are generally relatively far away from the attractors.

In sum, the results indicate that for $VP(1)$, public spaces directly connected to work or education facilities have the highest visit potential; for $VP(2)$, these spaces attached to attractors and nearby places of work or education, have the highest visit potential; and for $VP(3)$, public spaces with attractors and mainly located in the center of the district, have the highest visit potential.

Overall, the public spaces of schools, near *LIDL-headquarters* and those situated in *Cours du Commerce* and *Promenade Plantée* are potential public spaces. We can conclude that public spaces, close to population objects with high entrance/leaving rates, and at least connected to an attractor will have a high *VP* value.

### 5.2. Dynamic Visit Potential Model

In the second section of this analysis process, we are interested in the application of the dynamic Visit Potential Model, dynamically estimating the visit potential of the public space for each one-hour period between 06h00 and 23h00.

The analysis process is divided into two stages: first, the overall visit potential is calculated, which refers to the sum of all *VP* of public spaces in LaVallée; it is calculated for each time period *t* in the defined time-frame. Thereafter, we limit our analysis to only $VP(3)$, the most interesting variant. This choice is motivated by the good trade-off given by this variant between the proximity to people and the accessibility of attractors compared to the two first variants. In addition, the estimation of the pedestrian movement through public spaces is the main feature of this variant.

According to the results provided from the first stage, the three variants of *VPM* show very similar trends over the day. For instance, the observed tendencies of the third variant in the time-frame between 06h00 and 23h00 are detailed below.

From Figure 10 and at first glance, one might observe three peaks in the overall potential value of visits: in the morning around 8 h, followed by a peak at noon around 12 h, and finally in the evening around 17 h.

The variations of the global visit potential are presented as follows: first, LaVallée *VP* value is strongly increased during the morning peak between 06 h and 08 h and considerably declined between 8 h and 10 h. Next, different trends can be observed between 10 h and 14 h; initially, the *VP* value rises between 10 h and 12 h, followed by a slight increase until the midday peak from 12 h to 13 h, then decreasing thereafter. Furthermore, the results show that the overall *VP* is strongly increased between 15 h and 17 h, where it reaches its highest value during this evening peak. Then, it shows an opposite trend, it declines linearly from the beginning of the evening until the end of the day. Trends in visit potential in the three variants are presented in Figure 14.

The morning peak might be explained by the large flow of pedestrians moving through public spaces. This flow is generated by the important number of going-to-school and going-to-work trips. The students attend *Groupe Scolaire* and *Collège*, while almost all the employees work at *LIDL-headquarters*. A further explanation of these results can be given as follows. Based on the results shown in Figure 9, it can be seen that during this morning peak, approximately 40% of students are going to education places; 25% of the workers go to workplaces; and 20% of the LaVallée residents are moving through public spaces. Thus, approximately 2115 travelers are present in public spaces around 08 h. A visualization of the third dynamic visit potential at LaVallée during the morning period is shown in Figure 15.

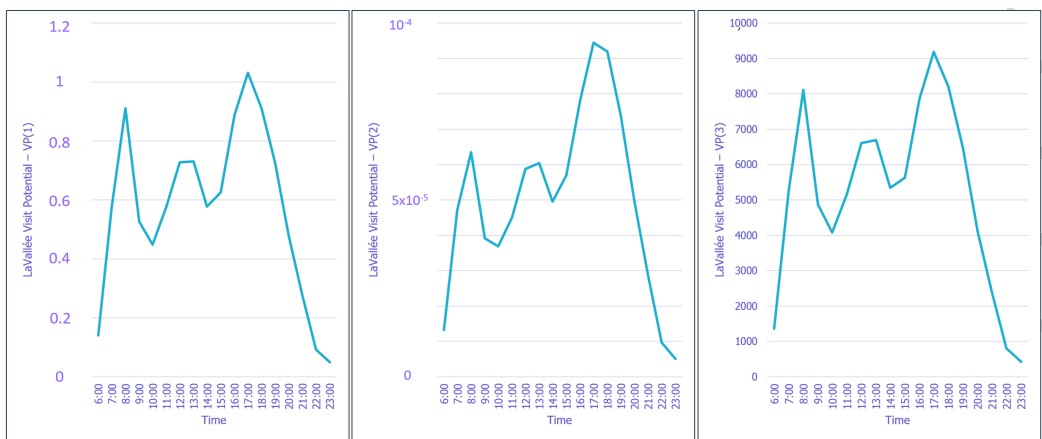

**Figure 14.** Trends in LaVallée's visit potential, for the three *VPM* variants. Three *VP* peaks are observed: morning around 8 h, midday, and evening around 17 h.

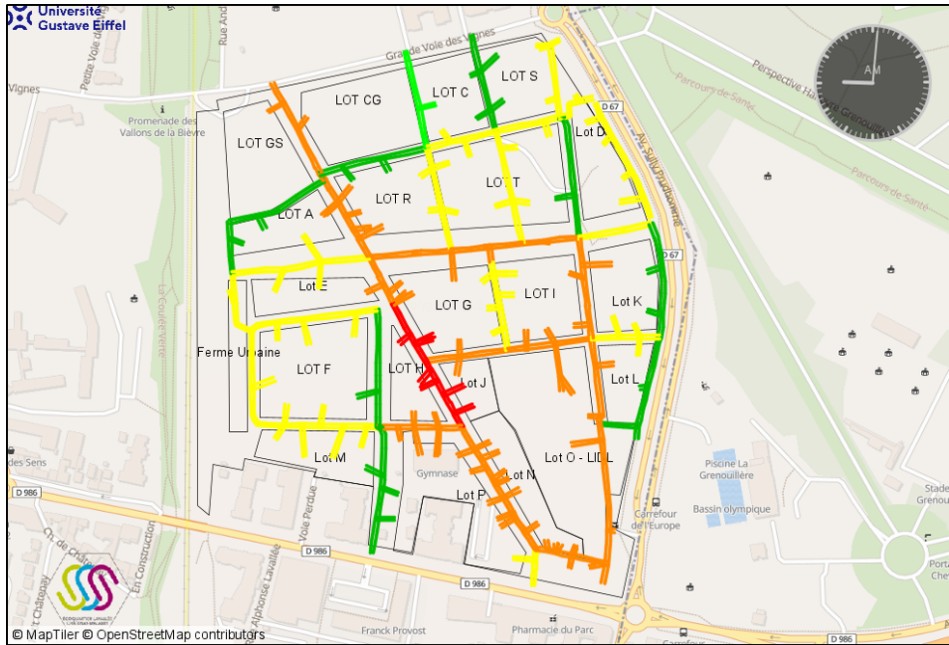

**Figure 15.** Dynamic Visit Potential as Aggregate Pedestrian Movement in LaVallée, during the morning period. Public spaces located on Cours du Commerce present a significant *VP* in the morning. Notably, the two related to Groupe Scolaire and Collège and to a lesser extent, those around LIDL.

The midday peak may be explained by the movements of the part-time workers and students in public spaces at noon (around 12 h). Moreover, it needs to be pointed out that at noon part of the population can carry out a secondary activity, in particular shopping. Thus, the pedestrian flow in public spaces nearby these facilities impacts the neighborhood *VP*. For instance, the public spaces situated in Cours du Commerce have significant potential at noon. In addition, based on the start/end rate of primary activities, the number of users moving through LaVallée during the midday peak is estimated at approximately 1760 people. An overview of the results of the third dynamic variant on the public spaces of LaVallée at noon is presented in Figure 16. One can see that people are engaging in activity of shopping to the LIDL grocery store and the commercial street and some are strolling through the Promenade plantée walking area.

At the evening peak, LaVallée recorded its highest *VP* value, which can be explained by the number of employees and students who are finishing their activities. In addition, an important proportion of LaVallée residents could participate in secondary activities, in

particular leisure and shopping. With regard to the start/end rates of primary activities given in Figure 9 and in contrast to the morning and midday peaks, it seems that trips to and from home, have an absolutely greater impact than the education and work together on the volume of flows in LaVallée. The number of travelers moving through public spaces is estimated at approximately 2512 agents. A glance about the third dynamic, in this evening period is given in Figure 17.

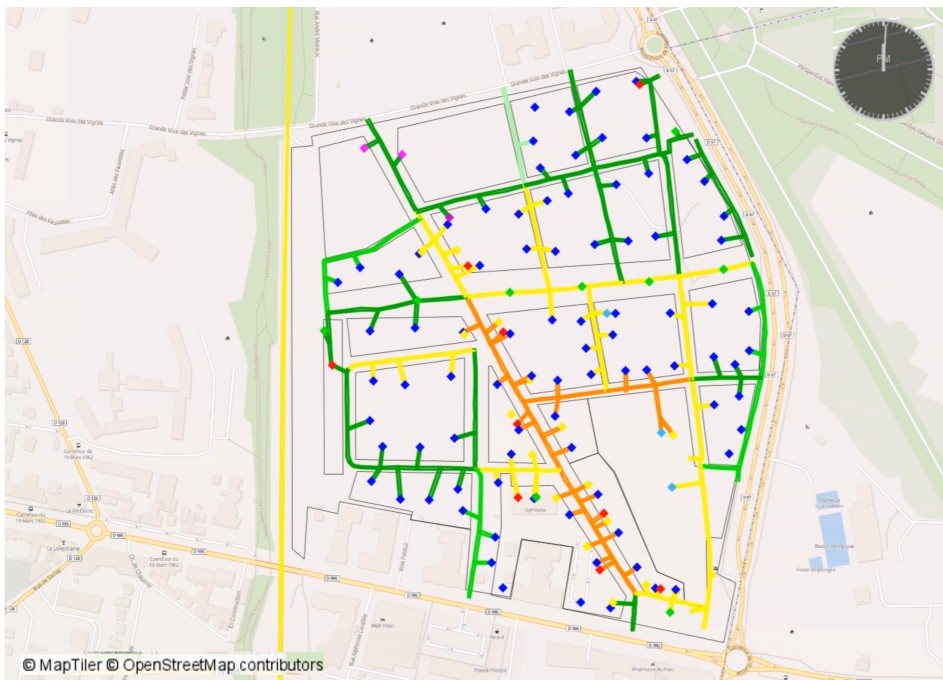

**Figure 16.** Dynamic Visit Potential as Aggregate Pedestrian Movement in LaVallée, during the midday period. LaVallée's public spaces have a lower *VP* value compared to the morning period, while *Cours du Commerce* and LIDL grocery store contains most of visit potential of the entire neighborhood.

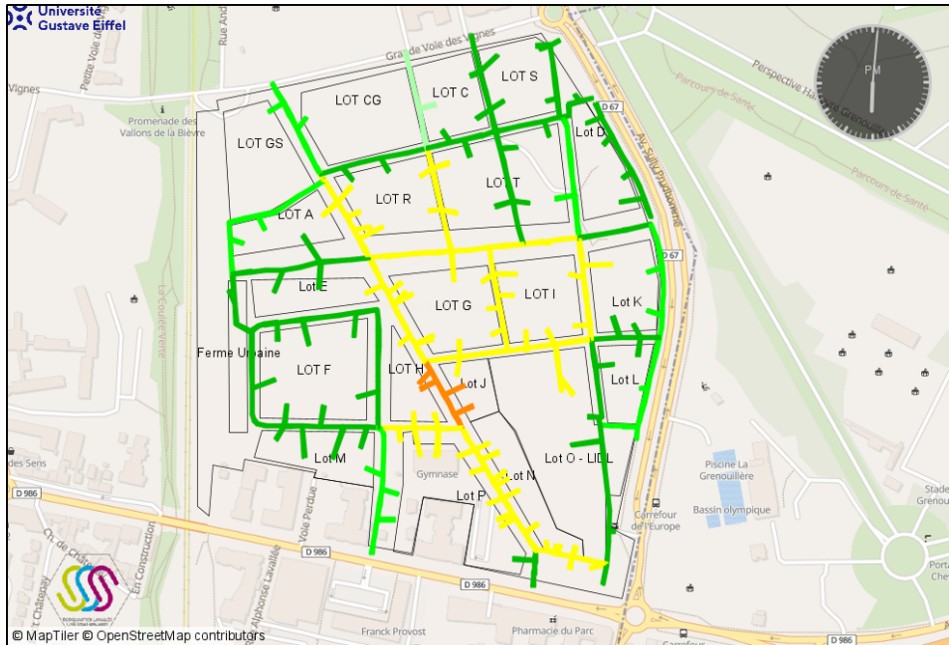

**Figure 17.** Dynamic Visit Potential as Aggregate Pedestrian Movement in LaVallée, during the evening period. LaVallée's public spaces have less potential in the evening than in the morning and at noon. The potential spaces are located around *Cours du Commerce* and *Promenade Plantée*, dense with shops, restaurants, and gardens.

In short, LaVallée's visit potential is mainly impacted by the entrance/leaving rates of population objects, the higher the rate, the higher the visit potential. From the results provided, it can be seen that there is a strong resemblance between the entrance/leaving rate of home activity and the trends of the overall *VP*.

*5.3. Validation of the Approach*

There are two aspects that need to be validated in the proposed method: the trip simulation over a day, with the daily planning of agents, and the urban hotspots resulting from the Visit Potential Model. The transportation simulation accuracy was previously evaluated in [44,48]. It provides result coherent with the travel census of the department of interest [43] in term of activity and travel mode. Especially the Enquête Global de Transport EGT2010 [45] that contains a detailed activity chain of a microsample of the Parisian population, and the ENTD data set [49] including the typical starting hour and duration distribution of activity by type. The marginals reported over the area of interest are respected: travel mode share and activity chain including work or education with secondary activities such as leisure, shopping, and restaurant. Joint attributes are also respected: the travel mode by type of activity is also respected (for instance 10% of trips for shopping are made by car). The time of engagement in an activity is conform to the census, both in start-time and duration of an activity. Furthermore, the simulation is coherent with a road traffic study realized by an external consultant during the preliminary impact study required to start the real-estate project (including only motorized vehicles traffic). The simulation was deemed correct according to the typical scientific literature criteria but the main focus of the paper is the hotspots prediction over the neighborhood; a recap of the obtained results is given in Figure 18. *dynVP* refers to the dynamic version of *VP*(3), the so-called dynamic VPM.

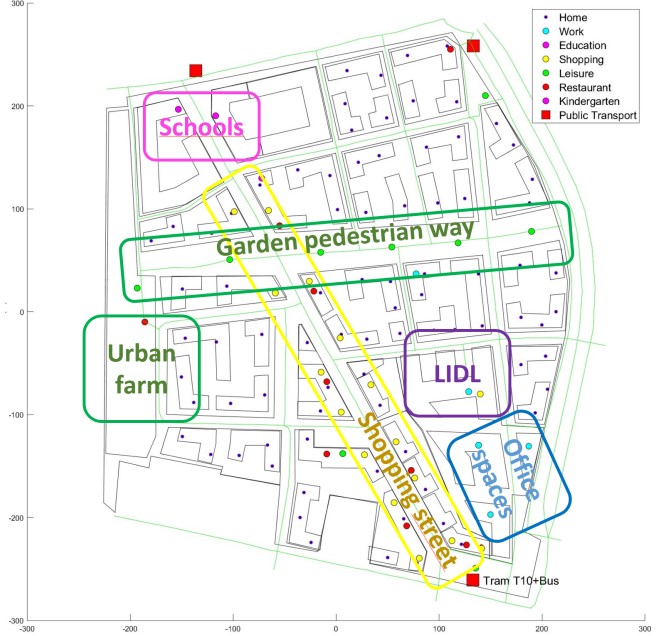

| Time | Method | Hotspots | Activity |
|------|--------|----------|----------|
| Day | VP(1) | Schools<br>Office spaces<br>Grocery store (LIDL) | Education<br>Work<br>Shopping |
| Day | VP(2) | Garden pedestrian way<br>Urban Farm<br>Shopping street | Leisure<br><br>Shopping |
| Day | VP(3) | Shopping street<br>Grocery store (LIDL)<br>Office spaces<br>Garden pedestrian way | Shopping<br><br>Work<br>Leisure |
| AM | dynVP | Schools<br>Office spaces<br>Shopping street<br>Grocery store (LIDL) | Education<br>Work<br>Shopping |
| Noon | dynVP | Shopping street<br>Grocery store (LIDL)<br>Garden pedestrian way | Shopping<br><br>Leisure |
| PM | dynVP | Shopping street<br>Grocery store (LIDL)<br>Garden pedestrian way | Shopping<br><br>Leisure |

**Figure 18.** (**Left**) the six polarities of LaVallée. (**Right**) main predicted hotspots for each method.

Most of the methods dealing with hotspot identification in the scientific literature are data-driven approach; they use measurement of people density (mainly from LBS data) and can assess the accuracy of their method on these data. For a district that does not exist yet, it is impossible to provide this type of accuracy measurement. However, it is possible to evaluate the soundness of the approach from a relative point of view of traditionally qualitative and intuitive realms of urban design. In the following, we will first note that the proposed method is coherent with common results of the literature on urban design.

Secondly, the method and its outputs was analyzed and commented by AEC (architecture, engineering, and construction) experts working on the real-estate project who expressed their prediction based on their empirical knowledge in urban planning.

To analyze the results, we will use the support of the illustration in Figure 18. It provides the six main polarities of the district according to the AEC experts. Indeed, during the planning phase of the real-estate project, the district was designed mainly around two streets: the shopping street across from north to south with a number of commercial facilities and a garden pedestrian way from east to west where no vehicles are allowed and a lot of vegetation and child playgrounds will be implanted. A small urban farm complements the garden pedestrian walkway at the west of the neighborhood. In the north are the two primary schools, while in the south are the office spaces hosting LIDL's future headquarters. Close to the office is the LIDL grocery store, a special kind of shopping facility as it is a large food retail area. The figure also illustrates the three public transportation access points, with the tramway/bus hub in the south and two bus stops in the north.

It is commonly known that the diversity of land use has a positive impact on urban vitality [1]. In [26], for instance, it was found that the primary influencing factor in determining hotspot locations was the presence of urban functions, such as activity facilities and notable public points of interest, which is generally being covered by our indexing of urban vitality. These factors hold more significance in explaining outcomes compared to building structures, accessibility, or human perception.

Density plays a pivotal role in enabling vibrant street life and providing accessible amenities, as highlighted in [50], which is promoted in LaVallée, while increasing the number of points of interest has some impact on enhancing neighborhood vibrancy; it is not as influential as encouraging a mix of complementary points of interest [35]. This is evident in Figures 15 to 17, where the intersection between the route to the grocery store and the central shopping street consistently emerges as the most crucial hotspot, regardless of the time of day.

Resident density, retail food density, and job density all contribute positively to urban vitality [51]. Factors such as proximity to parks/recreational areas, grocery stores, and bus stops, as well as non-residential zones (offices, retail spaces, institutions, and recreational areas) also play a significant role [52,53]. Specifically, the proximity to primary schools exerts a strong influence, indicating that primary schools often coexist with highly vibrant streets and are integral to creating a livable community space, as discussed in [38]. Together with functional diversity, they exert the most substantial influence on street vitality, which is evident in Figure 15, where the hotspot around the two schools is more pronounced.

The AEC experts working on the design of the real-estate project shared their experience in urban planning and shaded a light on the future of the neighborhood. According to them there are three kind of visitation of LaVallée to be expected, resulting in three categories of agents in our simulation:

- Residents: the prospective inhabitants of LaVallée district. At this stage of real estate development, their home locations remain uncertain as apartments are yet to be sold. Household compositions are assumed based on the number of flats per parcel. A total of 6200 residents are expected in 2027.
- Externals: These include workers and students who reside outside the district but work or study within LaVallée's workplaces or schools. There will be 1200 workers mostly at the LIDL headquarter offices and 1300 pupils.
- Visitors: Visitors come from outside the district and engage in shopping, dining, or other leisure activities within LaVallée during the day. A daily count of 2500 to 3000 visitors for leisure or shopping activities is to be expected according to our simulation.

The map of Figure 18 illustrates the location of the six polarities of the district. These are the main places around which the identity of the neighborhood has been built.

The shopping street is the main car road crossing the district: it is designed to be a place of high vitality similar to a shopping mall, which should attract a lot of external visitors. The garden pedestrian way join a large park—located on the outside east of the

district—and an urban farm. This pedestrian way will permit nice strolling and children playgrounds; it will act as a strong pull for leisure purposes. The urban farm is mostly reserved for the residents but it includes a restaurant that can be of interest to other visitors. The two schools are run by the city and the department to host pupils from around the location. They are a primary school and a junior high school for kids from 6 to 14 years old. The office spaces will be the headquarters of LIDL-France and in the long term will host all of the corporate staff of the retailer. Finally, a LIDL grocery store will be used by the residents and the people around the area.

The table of Figure 18 sums up the main prediction of the VPM methods. The dynamical version of the VPM, method dynVP, shows the various stage of the hotspots of the neighborhood during the day. It evolves as expected by the AEC with a lot of activity around the schools and the office spaces in the morning. Figure 15 shows that the schools and working places are active in the morning, along with the main street that will contain a lot of transit. People are engaging in shopping and leisure activities at noon and during the evening, around the main commercial street, the pedestrian way, and the grocery store. The grocery store is an active area in all periods of the day and especially after noon. The shopping street is expected to be the most important hotspot by the AEC experts as it is a transit road for the rest of the city, and will include a lot of the shopping stores and some food places, which is also the case in our forecast model.

## 6. Conclusions

This paper discusses the literature in the field of urban vitality and presents an adaptation of the Visit Potential Model for a case study of a future district still under construction. We gave an overview about the integrated Visit Potential Model proposed by [39], which was defined to evaluate public space characteristics. The model estimates the vitality of public spaces by their visit potential, focusing on human activity, particularly the pedestrian flows at the neighborhood scale.

The first part was devoted to the three variants of $VPM$, which estimate the visit potential as proximity to people denoted $VP(1)$, accessibility of attractors denoted $VP(2)$, and aggregate pedestrian movement denoted $VP(3)$. These three variants have been presented and, moreover, explicitly detailed. They are adapted to the case of simulated trips of people. Then, the given variants of $VPM$ were transformed into dynamics model, including the time dimension.

A case study of the LaVallée district was defined and carried out in two phases. In the first phase, the entrance/leaving rates and need frequency of the population objects and of the attractor objects were estimated from the data of LaVallée project and a public travel survey *EGT2010*. In the second phase, the dynamic entering/leaving rates were derived from the original rates and based on the results of a previous simulation by applying a specific process.

Finally, the integrated model was performed in the case of LaVallée in two stages: first, with a full time-frame between 06h00 and 23h00; second, using the dynamic models, with one-hour periods within the initial time horizon. The results obtained were analyzed and then discussed.

The results of the first series of variants confirmed the initial assumptions and showed remarkable conclusion about the potential of LaVallée's public spaces. Potential public spaces were located near schools and LIDL-headquarters for $VP(1)$; along Cours du Commerce and Promenade Plantée for $VP(2)$; and at LaVallée center for $VP(3)$.

Moreover, and applying the dynamics variants, the results obtained indicated that all variants followed the same trends, with three peaks: morning, midday, and evening. In addition, three peak periods of $VP(3)$ were explicitly analyzed: school public spaces have the highest $VP$ in the morning and those located in Cours du Commerce and La Promenade Plantée in the midday and the evening, when people are engaging in a shopping or leisure activity.

The existing literature on urban hotspot identification predominantly emphasizes refining the evaluation methodology, with limited attention given to its applicability in contexts such as formulating action plans for local urban renewal initiatives. As a result, it is not always clear to what extent these findings can be extrapolated to inform the design of other public spaces. This is especially pertinent for new designs without any existing human activity yet for assessment. A more comprehensive framework for quantitatively evaluating the qualities of public spaces is yet to be researched. Such a framework should be applicable to all components of a network of public spaces and capable of modeling the potential presence of people in these areas.

The proposed approach addressed the development of an urban characterization method based on activity-based travel models. We specifically focus on modeling multimodal travel at the city scale, particularly micro-mobility at the neighborhood level. This approach has an original application in urban planning, particularly within a mixed-use neighborhood construction project. It involves making predictions about the vibrancy of the future neighborhood solely based on territorial census data and upcoming activities on the site. This research responds to a growing academic interest in integrating systemic and scientific approaches into the traditionally qualitative and intuitive realms of urban design. We believe it will be valuable to the practitioners of the field, stakeholders, and urban planners.

**Author Contributions:** Investigation, Y.D. and R.B.; Methodology, Y.D. and R.B.; Supervision, R.B.; Writing—original draft, Y.D. and R.B.; Writing—review and editing,Y.D. and R.B. All authors have read and agreed to the published version of the manuscript.

**Funding:** This research was funded by the E3S project, a partnership between Eiffage and the I-SITE FUTURE consortium. FUTURE bénéficie d'une aide de l'État gérée par l'Agence Nationale de la Recherche (ANR) au titre du programme d'Investissements d'Avenir (référence ANR-16-IDEX-0003) en complément des apports des établissements et partenaires impliqués.

**Institutional Review Board Statement:** Not applicable.

**Informed Consent Statement:** Not applicable.

**Data Availability Statement:** Not applicable.

**Conflicts of Interest:** The authors declare no conflicts of interest.

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
