# Peer review of "Modeling Visit Potential to Predict Hotspots of a Future District"

_infrastructures, doi:10.3390/infrastructures8100145_

Round 1

Reviewer 1 Report

This is nice and well-prepared work. The study is presented clearly. The results are understandable and will be useful for other research. I formulated some smaller remarks to consider before the publication. This should increase the quality and readability of the paper.

Section 2 (related works) should end with summarizing today's state of the art. With pointing out the gaps in the research and the need for new findings. It should also the motivation (plan) of individual research presented in the manuscript and its goals.

Add a figure (map) with the location of the LaValle district in the Paris (or Ill de France) area.

Add a scale in all the maps (a line representing some distance for example 100 m).

No figures can be placed in the conclusion section or in the reference list. Figures 15 – 17 belong to section 5 and should be presented in this section. One possible solution is to minimize the size of these maps (to present two maps on one side). The second solution is a presentation of these maps in the appendix.

Some positions in the references should have more information allowing its identification and evaluation. For example, position number 16 (very important for study content) gives only the authors, title, and year, but no publisher, ISSN / ISBN, DOI, web address, etc.

Minor editing of English language required.

Reviewer 2 Report

The article is complete in all its parts, well-written, and worthy of being published.

A tip: in my opinion, the acronym VPM could be used for the term Visit Potential Model.

In my opinion, the study is not only scientifically sound but also useful to support the decisions to allow urban planners to optimize the allocation of resources and infrastructure development

Reviewer 3 Report

This article proposes a study of the Visit Potential Model, an integrated model for evaluating the characteristics of public spaces, which can help urban planners optimize the allocation of resources and infrastructure development. The model combines a universal law of visit frequencies in cities with a gravity measurement of accessibility and is represented as a graph by connecting public spaces to other spaces: population objects and attractor objects. The paper discusses the works cited in the field of urban vitality and presents an adaptation of the Visit Potential Model for a case study of a future district still under construction. The model estimates the vitality of public spaces as visit potential, focusing on human activity, particularly the pedestrian flows at the neighborhood scale.

The authors have implemented the Visit Potential Model developed by Herthogs et al. However, it is imperative for the authors to explicitly discuss the validation of this theoretical model against empirical (survey) data. A comprehensive and in-depth examination of the model's validation shall be included in the revised version of this study.

In the introduction section, when discussing the necessity of modeling visitation, we encourage the authors to incorporate references to case studies where they have previously endeavored to forecast area visitation for various applications as described in the manuscript.

Within Section 2 of the paper, it is recommended that the authors further delineate the research gap and underscore the originality and innovation inherent in their work.

Regarding Equation 1, How does the model predict entrance or departure rates?.

How does the model account for the potential presence of multiple attractors within the same proximal area?

Round 2

Reviewer 3 Report

The authors have made a significant effort in addressing the comments. Therefore I suggest the paper is published in the present form.

None